# Optimised nanobody-based quenchbodies for enhanced protein detection
Jordan H. Cater [1], Nehad S. El Salamouni [1,4], Ghada H. Mansour [2,4], Sebastian Hutchinson[2,4], Conall Mc Guinness [1], Stefan H. Mueller [1], Richard R. Spinks[1], Nirukshan Shanmugam[1], Adeline Pichard-Kostuch [2], Viktor Zahoransky[2], Harshad Ghodke[1], Marco Ribezzi-Crivellari[2], Haibo Yu [1,3], Antoine M. van Oijen [1] ✉, Andrew D. Griffiths [2] ✉ & Lisanne M. Spenkelink [1] ✉

Quenchbodies, antibodies labelled with fluorophores that increase in intensity upon antigen binding, offer great promise for biosensor development. Nanobody-based quenchbodies are particularly attractive due to their small size, ease of expression, high stability, rapid evolvability, and amenability to protein engineering. However, existing designs for protein detection show limited dynamic range, with fluorescence increases of only 1.1–1.4 fold. Here we identify the tryptophan residues in the nanobody complementarity-determining regions (CDRs) that are critical to quenchbody performance. Using a combination of rational design and molecular dynamics simulations, we developed an optimised nanobody scaffold with tryptophans introduced at key positions. We used this scaffold in an in vitro directed-evolution screen against human inflammatory cytokine interleukin-6 (IL-6). This yielded quenchbodies with 1.5–2.4-fold fluorescence increases, enabling IL-6 detection down to 1–2 nM. Our scaffold provides a valuable platform for developing biosensors for diverse protein targets, with applications in research, diagnostics, and environmental monitoring.

The detection of disease biomarkers is important in all aspects of modern medicine. Detection of proteins is particularly important because, in contrast to genetic markers, the presence of proteins is often connected with disease phenotypes directly[1]. Analytical mass spectrometry is the gold standard for the study of the proteome and identification of protein biomarkers[2–4]. However, mass spectrometry typically requires large and expensive equipment and involves lengthy and complex analysis for detection. In contrast, immunoassays are easy to use, rapid, selective, sensitive, and affordable. These features make immunoassays of high interest in clinical settings, where rapid diagnosis allows early treatment and prevention of the transmission of infectious diseases.

Immunoassays use antibodies for the detection of antigens. The readout in immunoassays can be based on a variety of output signals, including enzyme reaction monitoring, fluorescence, chemiluminescence, and chromatographic separation[5,6]. Regardless of their output signal, immunoassays can be generally divided into heterogeneous and homogeneous. In heterogeneous assays, the antibody-bound antigen requires separation from the unbound antibody and antigen. The enzyme-linked immunosorbent assay (ELISA) is a well-known example of a heterogeneous immunoassay relying on enzyme output that is often used in research and clinical settings[7]. Lateral flow assays are another type of common heterogeneous immunoassay that has become increasingly popular in recent years, especially in point-of-care settings, due to their speed and ease of use[8]. However, heterogeneous immunoassays typically require lengthy incubation times and extensive washes, making analyte detection laborious and time-consuming. Comparatively, homogenous "mix-and-read" immunoassays do not require a separation step and are, therefore, quicker and simpler. AlphaLISA (Perkin Elmer) is an example of a homogenous ELISA-style immunoassay that seeks to overcome laborious procedures associated with traditional ELISAs[9]. All these assays require a characterised pair of antibodies that can simultaneously bind the antigen at two different epitopes, which is considered the major bottleneck in the development of all sandwich-style immunoassays[8,10]. These issues can be overcome by the use of a fluorescently-labelled antibody, called a "quenchbody".

[1]Molecular Horizons, School of Chemistry and Molecular Bioscience, University of Wollongong, Wollongong, NSW, 2522, Australia. [2]Laboratoire de Biochimie, Chimie Biologie et Innovation, ESPCI Paris, Université PSL, Paris, France. [3]ARC Centre of Excellence in Quantum Biotechnology, University of Wollongong, Wollongong, NSW, 2522, Australia. [4]These authors contributed equally: Nehad S. El Salamouni, Ghada H. Mansour, Sebastian Hutchinson. ✉e-mail: vanoijen@uow.edu.au; andrew.griffiths@espci.fr; lisanne@uow.edu.au

Quenchbodies and their working principles are reviewed in detail elsewhere[11–13]. Briefly, quenchbodies are typically single-chain variable regions (scFv; 32 kDa) or antigen-binding fragments (Fab; 50 kDa) of antibodies, which have been labelled with a fluorophore via a flexible peptide linker. This flexible linker allows the fluorophore to interact with tryptophan residues in the quenchbody. This interaction leads to a small change in fluorescence polarisation ($1/r$) but a more marked reduction in fluorescence intensity (quenching) through photo-induced electron transfer[14] (Fig. 1A). The electron transfer is facilitated by hydrophobic/π–π stacking interactions, provided the fluorophore is within ≤10 Å of the quenching tryptophan[15–17]. The quenched fluorophore can increase in fluorescence intensity when binding of the quenchbody to its cognate antigen sterically hinders the fluorophore from interacting with the tryptophans (Fig. 1A). The increase in fluorescence intensity of the quenchbody upon antigen binding can be used as a homogenous immunoassay to rapidly measure concentration of antigens, even in complex biological samples such as human plasma[18,19].

Most current quenchbodies are based on scFv and Fab, but quenchbodies based on single-domain nanobodies derived from camelids are rare, despite nanobodies possessing several key advantages. These include small size (~15 kDa), ease of expression, high stability, evolvability, and amenability to protein engineering[20]. The animal-origin production of antibodies involves lengthy lead times and batch-based inconsistencies. In contrast, advancements in computational modelling[21] and synthetic nanobody production[22–25] have opened avenues for the rapid development of

quenchbodies that recognise a broad range of targets. Nanobody-based quenchbodies are particularly effective for recognition of smaller antigens like small-molecule drugs or peptides[26], with a maximal 6-fold increase in fluorescence intensity observed upon binding of a quenchbody to methotrexate[18]. However, recognition of larger protein antigens generally provides poorer responses[26], with fluorescent fold-increases of only 1.1–1.4 observed for albumin[11,19], claudin[27], human epidermal growth factor receptor-2[28], and hemagglutinin[29], possibly due to additional quenching caused by amino acids in these antigens. Optimisation of the position of key-nanobody tryptophans relative to the fluorophore is likely to yield improvements to the performance of nanobody-based quenchbodies.

We, therefore, sought to create a generalisable nanobody-based quenchbody scaffold containing tryptophans in optimal positions, which would be capable of recognising protein antigens with superior performance. We chose TAMRA conjugated via an N-terminal Cys-tag on the quenchbody with no linker, as this was identified to give the highest response in previous research, which screened multiple linker lengths and fluorophores for nanobody-based quenchbodies[18]. Structural-mechanistic in silico modelling of representative lysozyme-binding and maltose-binding protein (MBP)-binding nanobodies revealed the importance of intrinsic tryptophan locations. These quenchbodies were then subjected to rational mutagenesis to improve fluorophore-quenching. The resulting high-performing quenchbody scaffold was used as the basis for an in vitro directed-evolution screen against interleukin-6 (IL-6). The screen resulted in a nanobody-based quenchbody with a 2.4-fold increase in fluorescence

**Fig. 1 | Design and in silico modelling of de novo quenchbodies. A** Schematic representation of the working principle of a quenchbody. In the absence of the antigen (left), the fluorophore is quenched by tryptophans in the antibody. In the antigen-bound state (right,) the fluorophore is displaced, resulting in dequenching and an increase in fluorescence intensity. **B** Proposed mechanism of the MBP-binding quenchbody (blue) modelled from PDB ID: 5M14, with N-terminal covalently conjugated fluorophore (green) undergoing quenching due to interaction with the intrinsic CDR-based tryptophans (red spheres). Upon binding to the MBP antigen (grey surface model), the fluorophore is sterically occluded from tryptophans (W101, W110 and W115), which is associated with increased fluorescence intensity. **C** Average normalised distribution histogram of the distance of the fluorophore to any of the three tryptophans (W101, W110 and W115) derived from MD simulations in the absence (blue) or presence (green) of antigen for the MBP-binding nanobody. The fluorophore is considered quenched by tryptophan at distances ≤10 Å (hatched zone). Individual plots for W101, W110 and W115 are provided (Fig. S1A). **D** Average normalised distribution histogram of the distance of the fluorophore to either of the two tryptophans (W103 and W115) derived from MD simulations in the absence (blue) or presence (green) of antigen for the lysozyme-binding nanobody. Individual plots for W103 and W115 are provided (Fig. S1B).

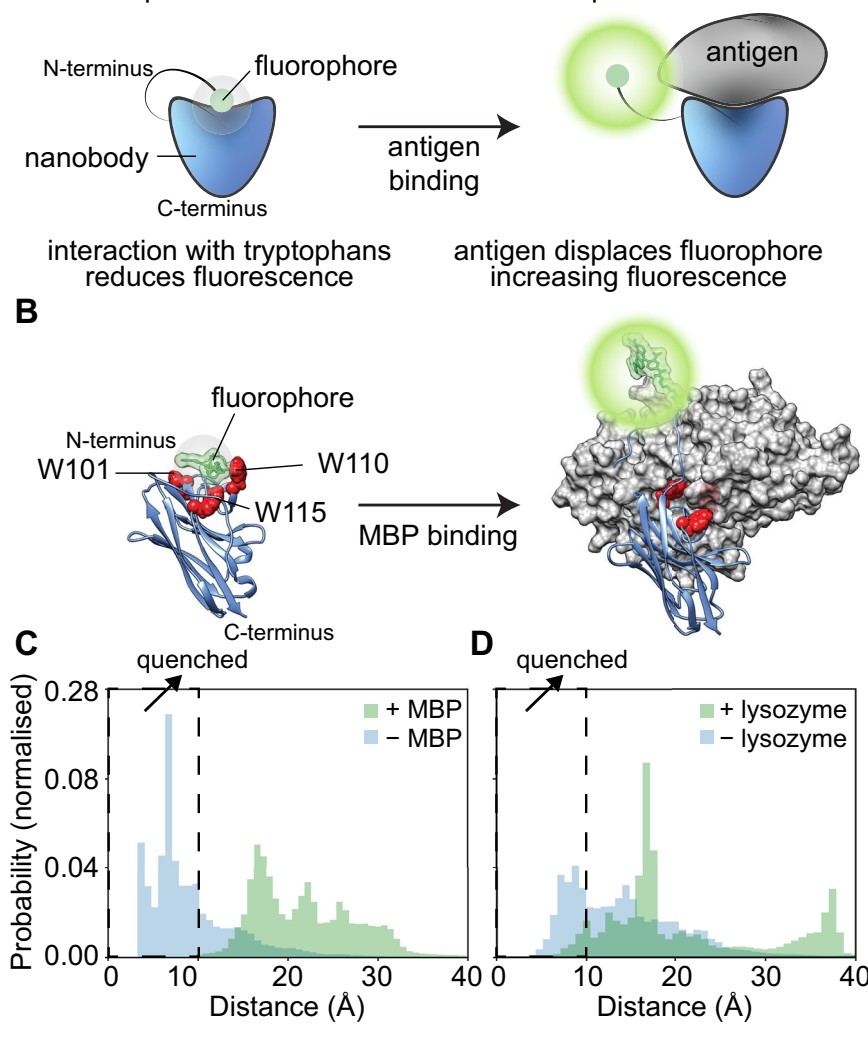

**Fig. 2 | Fluorescence intensity changes in quenchbodies upon antigen binding. A** MBP or (**B**) lysozyme quenchbodies were incubated (60 min, 25 °C) in the presence of increasing concentrations of their cognate antigens. Data are mean ± SD of the fluorescence fold-increase relative to the 0 nM antigen native sample (excitation = 535 ± 20 nm and emission = 585 ± 30 nm). The lowest point of statistically significant detection is indicated (Ordinary one-way ANOVA, Tukey's multiple comparisons, $p = 0.002$), with concentrations listed to the left of the hashed line considered to be non-significant ($\alpha = 0.05$). Fluorescence intensity responses of (**C**) MBP or (**D**) lysozyme quenchbodies to their cognate antigens were quantified and fitted to an equation describing a single site-specific binding mode to derive an $EC_{50}$ as a proxy measure for quenchbody binding affinity ($K_D$). Data are mean ± SD normalised fluorescence intensity.

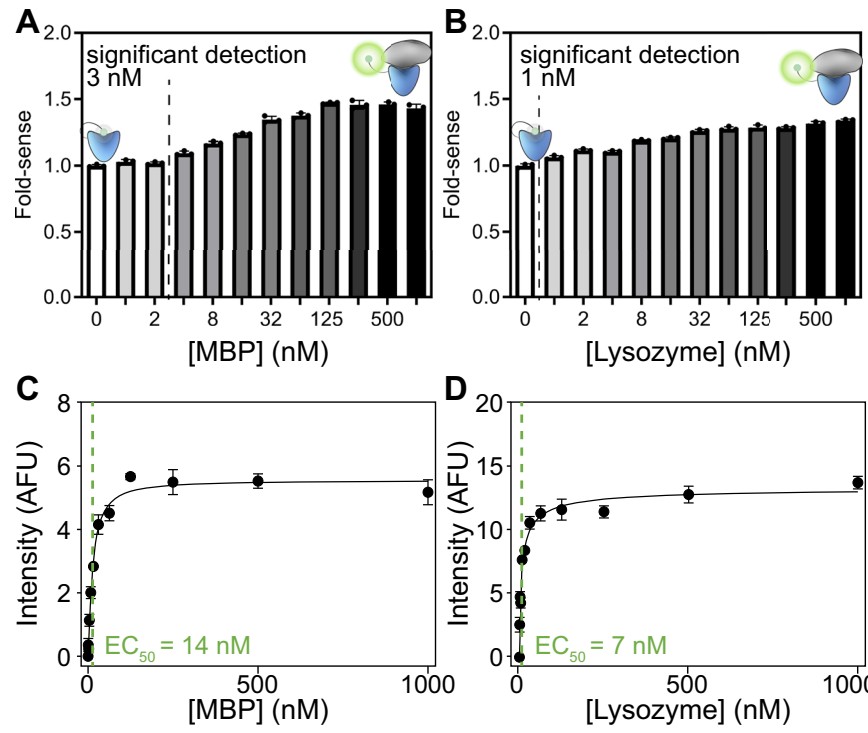

intensity when detecting IL-6. Our high-performance nanobody-based quenchbody is a generalisable scaffold suited to the production of fluorescent biosensors for the detection of a variety of clinical protein antigens.

## Results

### In silico evaluation of de novo quenchbody models

To develop a nanobody-based quenchbody scaffold with high performance (i.e., displays a large fluorescence increase upon antigen binding), we first considered the binding mode of nanobodies with their antigens. It has been shown previously that nanobodies tend to prefer three distinct binding modes known as concave, convex or loop, named for the shape of the nanobody antigen-binding interface[30]. Concave-binding nanobodies might preclude fluorophore access, while loop-binding nanobodies are considered too mobile, unstructured or elongated for consistent fluorophore interaction[31]. Given the favourable compact interface available to the N-terminal fluorophore of convex-binding nanobodies, we decided to exclusively use these. Using this constraint, we chose convex-binding nanobodies against maltose-binding protein (MBP) (PDB ID: 5M14) and lysozyme (PDB ID: 1ZVH) as initial scaffolds, because of the availability of high-quality (<2 Å) crystal structures in complex with their antigens[30,32]. We carried out all-atom molecular dynamics (MD) simulations in the absence and presence of antigen. We then calculated the distances of the conjugated fluorophore to each of the intrinsic tryptophan, as photo-induced electron transfer is believed to occur at distances ≤10 Å[15,16,33,34]. The W36 framework residue conserved in both nanobodies is deeply buried within the interior of the nanobody and was therefore excluded from analysis. For the MBP-binding nanobody, which contains three intrinsic tryptophans in the CDRs (Fig. 1B), fluorophore–tryptophan distances of ≤10 Å were observed in the unbound state but not the antigen-bound state (Fig. 1C). Notably, distances of ≤10 Å were mostly observed for W110 and W115 (Fig. S1A) and less with W101. The average simulation time the fluorophore spent at distances ≤10 Å from any of the three intrinsic tryptophans was 70.7 ± 18.7% in the unbound state and 0.1 ± 0.2% in the antigen-bound state (Fig. 1C; Table S1). Comparatively, the lysozyme-binding nanobody had two tryptophans in the CDRs (W103 and W115) that could act as potential quenchers. Fluorophore–tryptophan distances of ≤10 Å were observed for both the unbound and antigen-bound states (Fig. 1D). Distances of ≤10 Å were more

frequently observed in the antigen-bound state for W115 but not W103 (Fig. S1B). Correspondingly, the average proportion of time the fluorophore spent ≤10 Å from any of the tryptophans in the lysozyme-binding nanobody was 37.3 ± 24.1% in the unbound state and 5.9 ± 6.1% in the antigen-bound state (Table S2).

### In vitro production and biochemical characterisation of quenchbodies

Now that we had identified putative quenching tryptophans in both quenchbodies, we wanted to confirm these experimentally. We developed a cell-free expression protocol that allows for the expression, labelling, and purification of quenchbodies within 24 h. Sequences encoding the nanobodies were grafted into an expression cassette in silico. The scaffold contains a Cys-tag for N-terminal conjugation of the fluorophore[18,19], a C-terminal Avi-tag for biotinylation, and a 3× FLAG tag for purification (Fig. S2A, B). Following cell-free protein expression, quenchbodies were captured using anti-FLAG magnetic beads, labelled with the fluorophore TAMRA, biotinylated, extensively washed, and finally eluted using excess FLAG peptide. This workflow typically resulted in 10–50 μg of quenchbody. Purified quenchbodies showed a single fluorescent band (Fig. S2C,D), supporting successful labelling of quenchbodies. Spectrophotometric analysis of the quenchbody suggested that the labelling efficiency was approximately 100%. To test the antigen-binding activity of the quenchbodies, we carried out an antigen pull-down assay (see methods). SDS-PAGE analysis showed bands consistent with the size of the antigens MBP (~43 kDa) and lysozyme (~15 kDa) alongside the expected size for each of the quenchbody bands (~21 kDa) (Fig. S2C, D).

To test the performance of the quenchbody, we used a 384-well plate fluorescence assay. We measured the increase in fluorescence in the presence of antigen, compared to the intensity in the absence of antigen (Fig. 2). We use the ratio of the two intensities to define the fold-sense—the performance of the quenchbody. Qb-MBP showed a maximal 1.5 fold-sense at ≥100 nM MBP, with statistically significant detection of as low as 4 nM MBP (Tukey's multiple comparisons, $p = 0.0002$) (Fig. 2A). In line with our in silico predictions, a slightly lower 1.3 fold-sense was observed for Qb-Lys at ≥1000 nM lysozyme, with statistically significant detection of as low as 1 nM lysozyme (Tukey's multiple comparisons,

$p = 0.0002$) (Fig. 2B). The half-maximum effective concentration (EC$_{50}$) was calculated to be 14 nM for Qb-MBP (Fig. 2C) and 7 nM for Qb-Lys (Fig. 2D), which is comparable with previously-reported binding affinities for the 5M14 MBP-binding nanobody ($K_D = 24$ nM)[30] and the 1ZVH lysozyme-binding nanobody ($K_D = 60$ nM)[32], respectively. The similarity of these values suggests that the fluorophore on the quenchbody does not significantly affect the antigen-binding capacity of the nanobody, and that the fluorescence assay is accurately reporting the fraction of antigen bound.

## CDR-tryptophans underpin the quenchbody sensing mechanism

Photo-induced electron transfer from intrinsic quenchbody tryptophans to the fluorophore has previously been identified as the key quenching mechanism[11–13]. However, there are no clear guidelines for where tryptophans are best placed in nanobody-based quenchbodies, especially when aiming for generalisable detection of different antigens. There are three native tryptophans in the lysozyme quenchbody, at positions 36, 103, and 115. Of these three, only W103 and W115 are in the CDRs. W103 appears to interface directly with lysozyme, while W115 is on the surface of Qb-Lys but does not interface with lysozyme (Fig. 3A). The third tryptophan, W36, is buried deeply within the beta-sheet-rich barrel of the nanobody and is therefore unlikely to quench the fluorophore (Table S3).

To test the effect of the putative quenching tryptophans in the lysozyme quenchbody, we targeted W103 and W115 for substitution. We substituted with tyrosines to mimic the biochemical properties of tryptophan residues to minimally disturb the quenchbody structure and function in accordance with FoldX predictions (Table S4)[35]. We created Qb-Lys variants W103Y, W115Y, and W103Y/W115Y and expressed these using our cell-free expression method (see methods). Pulldown assays showed that W115Y had equivalent antigen-binding activity compared to WT, while W103Y showed slightly reduced binding (Fig. S3). This reduction was expected given that the W103 residue is directly involved in lysozyme binding. Next, we tested their performance using the fluorescence plate-reader assay (see methods).

In the presence of 500 nM lysozyme, the W115Y mutant had near identical performance to the WT control, indicating that this tryptophan was unlikely to be important in the quenching mechanism (Fig. 3B). This is consistent with our MD simulations, where W115 was accessible to the fluorophore in the lysozyme-unbound and bound states (Fig. S1B). In contrast, there was a significant reduction in fold-sense for the W103Y mutant, compared to the WT (Fig. 3B), suggesting an important role for this tryptophan in the quenching mechanism. Taken together, these data suggest that CDR-based tryptophans that directly interface with antigens are the most important for the performance of quenchbodies.

Based on this information, we subsequently attempted to improve the performance of the lysozyme quenchbody by reverse-substituting CDR-tyrosines with tryptophans (Y27W, Y104W and Y110W). MD simulations showed that Y27W, Y104W and Y110W could each contribute to the quenching of the fluorophore in the unbound state for ~50% of the time but for only ~5% of the time in the antigen-bound state (Fig. S4 and Table S5). FoldX predicted Y27W, Y104W and Y110W would be suitable for maintaining protein stability and antigen binding (Table S6)[35]. We produced single, pairwise, and triple Y27W, Y104W and Y110W mutants of the quenchbody (Fig. 3C). All of these mutants had similar antigen-binding capacity as the WT (Fig. S5). Interestingly, the Y110W mutant and the pairwise mutants that include Y110W showed a marked improvement in fold-sense. (Fig. 3C). The Y110W single mutation emerged as the best for maximising fold-sense (Y110W = 1.7, compared to WT = 1.3), while single mutations of Y27W or Y104W were equivalent to WT (Fig. 3C).

To test the generalisability of this strategy to other quenchbodies, we next targeted the MBP quenchbody. First, we substituted intrinsic CDR tryptophans which contact the antigen W101, W110 and W115 (Fig. 3D) for alanines to confirm the tryptophans most important for quenching. Knockout of W101, W110 and W115 showed no response, supporting the importance of these tryptophans (Fig. 3E). We next reverse-substituted CDR-tyrosines that contact the antigen with tryptophans, aiming to improve the quenchbody performance, similar to earlier

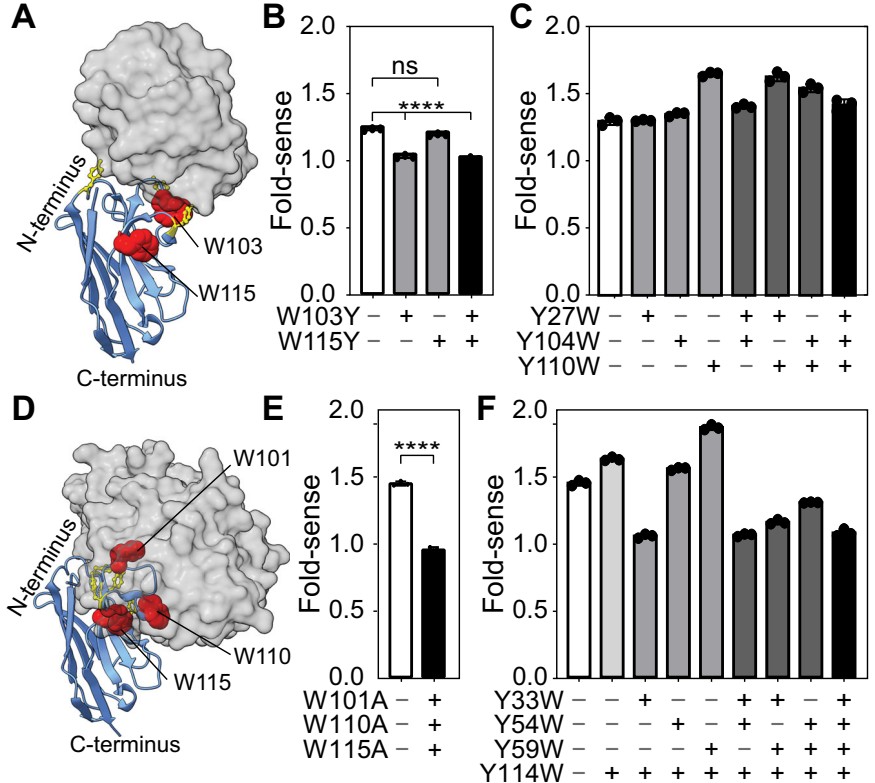

**Fig. 3 | Tryptophan-mediated quenching in lysozyme and MBP quenchbodies. A** Model of the lysozyme quenchbody (blue) highlighting all intrinsic tyrosines (yellow sticks) and tryptophans (red spheres, as labelled), in complex with lysozyme (grey surface model), based on PDB ID: 1ZVH. **B** Fold-sense for WT, W103Y, W115Y and W103Y/W115Y lysozyme quenchbody variants in the presence of 500 nM lysozyme. **C** Fold-sense for WT, Y27W, Y104W, Y110W, single-, double- and triple-mutant lysozyme quenchbody variants. **D** Model of the MBP quenchbody (blue) highlighting all intrinsic tyrosines (yellow sticks) and tryptophans (red spheres, as labelled), in complex with MBP (grey surface model), based on PDB ID: 5M14. **E** Fold-sense for WT, vs W101A/W110A/W115A triple mutant in the presence of 500 nM MBP. **F** Fold-sense for WT, Y33W, Y54W, Y59W and Y114W, double-, triple-, and quadruple-mutant MBP quenchbody variants. Data are mean ± SD (excitation = 535 ± 20 nm and emission = 585 ± 30 nm). Ordinary one-way ANOVA with Tukey's multiple comparisons shows significant (**** = $p < 0.0001$) or non-significant (ns) differences.

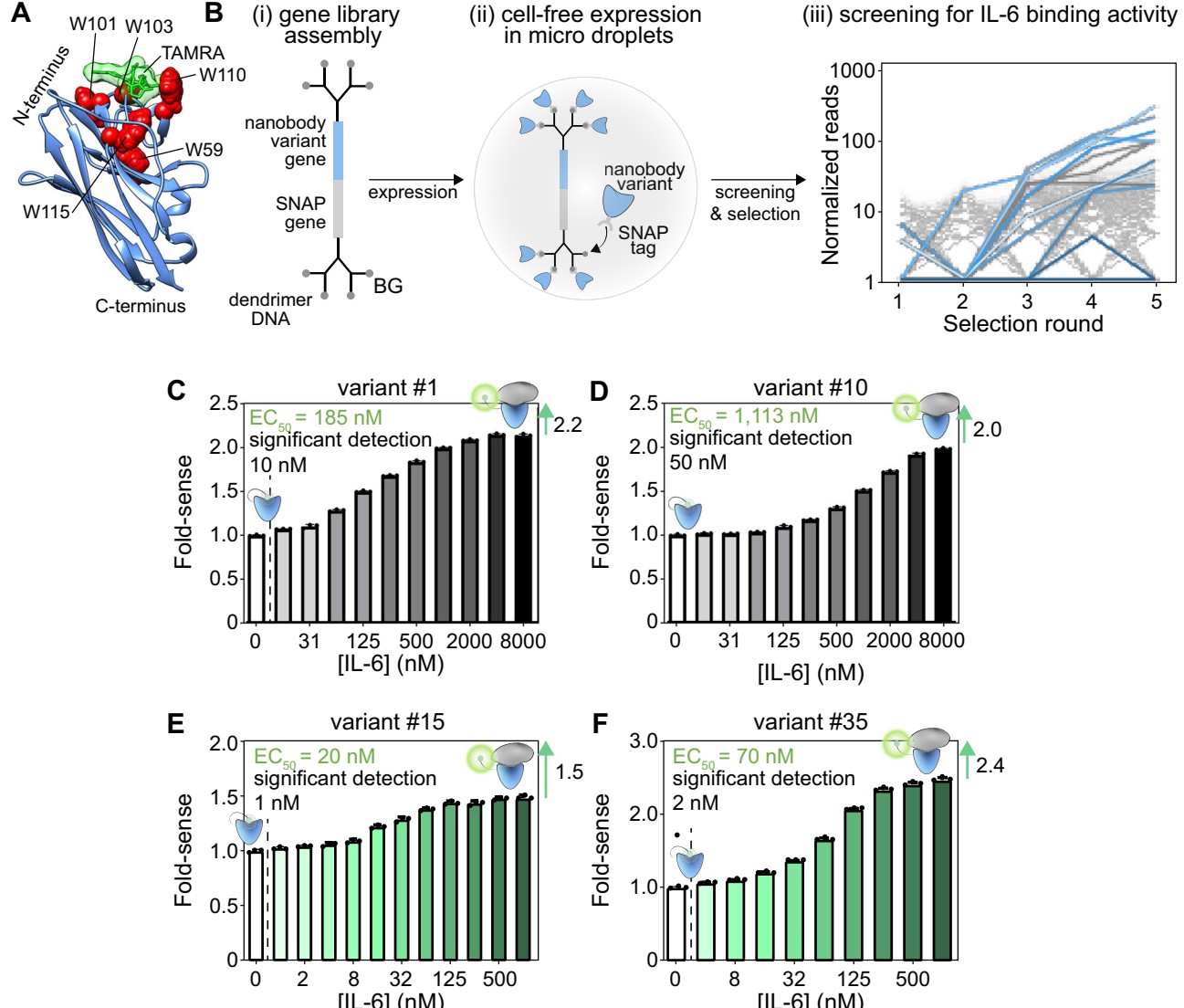

**Fig. 4 | Evolution of novel CDR-tryptophan quenchbodies for binding to interleukin-6. A** Model of our tryptophan-optimised nanobody scaffold. **B** Schematic representation of the directed-evolution workflow. (i) A gene library consisting of variant genes (blue) linked to SNAP genes (grey) is assembled and ligated to dendrimer-like DNA conjugated to a benzylguanine moiety (BG). (ii) Gene constructs are encapsulated in individual microdroplets with cell-free expression reagent. Expressed SNAP-tagged variants bind to the gene construct, resulting in phenotype–genotype linkage. (iii) Variants are screened and sequenced to measure the number of reads for each variant. Lines show normalised read counts over rounds 1–5, selecting for IL-6 binding over 2 technical replicates. Each hit is indicated as a separate colour. **C** The fold-sense for quenchbody variant #1 and **D** variant #10 in the presence of increasing concentrations of IL-6. The lowest point of statistically significant detection is indicated (Ordinary one-way ANOVA, Tukey's multiple comparisons, $p < 0.0001$), with concentrations listed to the left of the hashed line considered to be non-significant ($\alpha = 0.05$). **E** The fold-sense for affinity-matured quenchbody variant #15 in the presence of increasing concentrations of IL-6 shows an $EC_{50} = 20$ nM, and significant detection down to 1 nM. **F** The fold-sense for affinity-matured quenchbody variant #35 in the presence of increasing concentrations of IL-6 shows a fold-sense of 2.4 and significant detection down to 2 nM.

experiments conducted on the lysozyme quenchbody. A Y114W single mutant was first produced as it was considered potentially redundant based on its proximity to existing W110 and W115. The Y114W mutant retained its binding affinity and slightly improved the quenchbody fold-sense to 1.6 (the WT is 1.5). Therefore, the Y114W mutant was kept constant in further single, pairwise and triple mutations. The Y59W/Y114W mutant had an improved fold-sense of 1.9, suggesting a large improvement due to Y59W (Fig. 3F). For all other mutants, fold-sense was lower than the Y114W mutant, and in some cases lower than WT (Fig. 3C), which was possibly a result of complete loss of binding to the antigen as observed by pulldown assay (Fig. S6). Formally, our experimental results do not disentangle loss of antigen binding and loss of quenching. Instead, our results report on the overall quenchbody performance, which depends on both antigen binding and quenching by the

tryptophans, with the relative importance of each tryptophan further considered using complementary in silico modelling. Based on this information, tryptophans that contributed favourably to sensing performance across the lysozyme and MBP quenchbodies were considered to be W59, W101, W103, W110 and W115.

## In vitro evolution of CDR-tryptophan quenchbodies for IL-6 detection

Using this information on the key tryptophan positions, we generated an optimised quenchbody scaffold. This scaffold has tryptophans in position 59 in CDR2, and positions 101, 103, 110 and 115 in CDR3, forming a convex multi-tryptophan surface on the nanobody (Fig. 4A). We then used this scaffold as the basis for a directed-evolution study, aiming to generate a high-performing quenchbody against a physiologically-

relevant protein target, interleukin-6 (IL-6). We first assembled a synthetic nanobody gene library in which the CDRs were randomised by TRIM oligonucleotides, in line with a previous strategy (Fig. S7)[30]. All variant genes were attached to a SNAP gene, separated by a G/S-rich linker, and dendrimer-like DNA conjugated to benzylguanine was ligated to the expression cassette (Fig. 4B, left). The gene library was encapsulated in water-in-oil emulsions with a cell-free protein expression reagent. We ensured that we encapsulated a maximum of one variant per droplet. Once expressed, the SNAP tag will form a covalent bond with the benzylguanine on the gene construct, resulting in linkage of protein phenotype to genotype (Fig. 4B, middle)[36,37]. After protein expression, the emulsion was broken, and the variants were selected against magnetic beads displaying IL-6. The selected DNA was recovered by PCR, and sequences were tracked by Illumina NGS on a MiSeq instrument (Fig. 4B, right). The number of reads that passed quality filtering (see methods) increased from 59% in round 1 to an average of 71% (± 2.6% SD) by round 5 (Fig. S8A–C), indicating a purifying selection on correctly assembled nanobody genes. Finally, we extracted the top twelve hits from these screens based on their abundance in round 5 (Fig. S8D–F).

The sequences for these hits were each grafted into the cell-free-expression cassette and expressed, labelled, and purified in vitro (Fig. S9A, B). Screening of these IL-6 quenchbodies in a fluorescence plate assay showed sensing by three out of twelve responders (Fig. S9C). Further characterisation indicated a maximal 2.2-fold increase for quenchbody #1 at ≥4000 nM IL-6, with $EC_{50}$ = 185 nM (Fig. 4C). A maximal 2.0-fold increase in fluorescence intensity was observed for quenchbody #10 at ≥8000 nM IL-6, with $EC_{50}$ = 1113 nM (Fig. 4D).

We sought to increase the affinity of IL-6 binding quenchbodies by introducing single randomised codons into each CDRs 1–3 for Qb-IL6-1, 2 and 10 (Fig. S10A). Furthermore, we constructed a library from oligopools, in which all the CDRs from the functional quenchbodies were mixed (Fig. S10B, D). We carried out three rounds of selection and tracked selections by Illumina NGS on a MiSeq instrument (Fig. S10C, Fig. S11A). Enrichment scores were determined by dividing the relative abundance in round three by the abundance of blank bead controls. Hits were selected by filtering for sequences with an enrichment score >200 across IL-6 selections (see methods). From this, we identified 24 hits. We again grafted the gene sequences for these hits into the cell-free-expression cassette for production in vitro (Fig. S11B, C).

Remarkably, screening of these quenchbodies in our fluorescence plate-reader assay revealed eleven high-performing quenchbodies (Fig. S11D, Fig S12). In particular, variant #15 had an $EC_{50}$ of 20 nM with significant detection down to 1 nM (Fig. 4E), while variant #35 had a fold-sense of 2.4 with significant detection down to 2 nM (Fig. 4F), together epresenting a 1.6–2-fold increase in fold-sense compared to parent quenchbodies. In some cases, lack of fold-sense response from quenchbodies was observed to correlate with poor nanobody expression or structural heterogeneity (Fig. S11B, C). Addition of an in-solution quencher (100 mM potassium iodide) did not change the fluorescence response of the null-responder (variant #27) and slightly reduced the response of a low-responder (variant #15) (Fig. S13), suggesting the fluorophore in the antigen-bound state is predominantly exposed to solvent. Substitution of favourable tryptophans with tyrosines, to produce W59Y/W101Y/ W103Y/W110Y/W115Y combinatorial mutants, showed a complete lack of fluorescence response to IL-6 in variants #13, #15 and #36 (Fig. S14), supporting the importance of these tryptophan positions for quenchbody performance. We chose variants 15, 33 and 36 for specificity testing, which showed similar but slightly reduced fluorescence increases in the presence of IL-6 after spiking with 50% plasma (Fig. S15), which may reflect non-specific quenching of TAMRA by high concentrations of plasma proteins. Taken together, the data show that our CDR-tryptophan-optimised synthetic nanobody scaffold is generalisable and can be used to generate specific quenchbodies with high sensing performance when detecting future protein targets of interest.

## Discussion

Through a combination of in silico modelling, biochemical characterisation, and targeted mutational screening, we developed an optimised nanobody-based quenchbody scaffold that can be used to rapidly produce quenchbodies with superior sensing performance for proteins, which are challenging targets for quenchbodies, given that proteins contain amino acids that can quench the fluorophore, impeding quenchbody performance. Using this scaffold in an in vitro evolution screen, we were able to generate novel quenchbodies for the detection of IL-6 with 1.5–2.4 fold-sense. This is a marked improvement over the fold-sense of 1.3–1.5 obtained for the parental quenchbodies we generated as models against MBP and lysozyme in the present study. While higher fluorescence fold-increases have been reported for larger antibody- or ScFV-based quenchbodies[11,27–29,38], or for nanobody-based quenchbodies that detect small molecules such as methotrexate[18,19], other nanobody-based quenchbodies that detect purified proteins only report fluorescence fold-increases of 1.2–1.4[19,39]. In addition, the present scaffold may hold superior detection performance for non-protein-based antigens, such as small molecules, metabolites or other biomolecules which lack quenching amino acids.

Our tryptophan-optimised quenchbody is nanobody-based, which holds several advantages over more common scFv-based or Fab-based quenchbodies. These advantages include higher stability, tolerance to mutation, and ease of production[12,20]. This work therefore demonstrates a generalisable strategy for the rapid production of nanobody-based quenchbody biosensors with high fluorescence performance against proteins of interest within 2–3 weeks. Another advantage of the present design is that nanobody-based sensors can be produced entirely synthetically, eliminating the need for animal or cell-based platforms, which typically require high amounts of antigen, as well as requiring that the antigen is not toxic or disruptive to the organism's normal functions. Given that we were able to successfully evolve high-performing quenchbodies against IL-6, a master mediator of human inflammation[40], these tryptophan-optimised quenchbodies hold great promise for clinical, diagnostic, or therapeutic applications, such as the detection of IL-6 and potentially other biomarkers in biological fluids.

This work also provided unique insights into the quenching mechanism involving key tryptophan residues of nanobody-based quenchbodies. In scFv and Fab-based quenchbodies, semi-conserved tryptophan residues in the VH–VL interface (not in the CDRs), quench the flexible N-terminal fluorophore in the unbound state, with antigen binding causing allosteric displacement of the fluorophore from the interface through a conformational change in the scFv[11–13,41]. Our study supports a different mechanism for nanobody-based quenchbodies, whereby CDR-tryptophan residues that directly interface with the protein antigen are the most important for fluorophore quenching. This mechanism is similar to previous tryptophan-substitution experiments involving a nanobody-based quenchbody for methotrexate. These experiments revealed the methotrexate-contacting W34 in CDR1 as a key quenching residue for methotrexate detection[18]. However, in the latter case, methotrexate binding translocated W34 of CDR1 away from the fluorophore via a conformational change in the antigen-bound state[18]. In contrast, our quenchbody scaffold contains a convex-binding surface for generalised dequenching in the antigen-bound state. As such, CDR1–W34 quenchbodies may be more useful for recognising smaller antigens (haptens) which cause molecular displacement of CDR1, whereas our convex-binding quenchbodies may be more powerful for recognising larger antigens, such as proteins, which block the convex surface formed by CDR1, 2 and 3.

While we were able to generate high-performing IL-6 quenchbodies, poorly-performing quenchbodies were also obtained. High-resolution experimental structures of the quenchbody-IL6 complexes will likely be key for determining why some quenchbodies were good responders, whereas others failed to respond. We speculate that some IL-6 quenchbodies failed to respond or had lower responses, because IL-6 is a relatively small antigen, and depending on the mode of binding, it is possible that

tryptophans are accessible to the fluorophore at the antigen–quenchbody interface if IL-6 does not completely envelope the convex-binding surface of the quenchbodies. Therefore, some tryptophans may be redundant for successful performance, and higher quenchbody performance may be possible through further engineering of library-derived quenchbodies to remove redundant tryptophans. Alternatively, the evolution of convex-binding nanobodies containing conserved CDR-tryptophans to bind large protein targets which completely occupy the convex-binding surface might lead to higher fluorescence performance for future quenchbody development. It is also possible that the bound antigen may quench the N-terminal fluorophore of the quenchbody, given its proximity, especially if the bound antigen has surface-exposed amino acids capable of quenching the fluorophore. The latter may in part explain why the fluorescence fold increase (fold-sense) observed for quenchbodies binding proteins is generally lower (1.1–1.4-fold)[11,19,27–29], compared to some extreme cases observed for the fluorescence fold-increase of quenchbodies against peptides (<9.6-fold)[42,43] and small molecules (<6.0-fold)[18], which may lack quenching amino acids such as tryptophans. Although quenchbodies currently perform well for the detection of smaller molecular antigens, protein antigens remain a challenging but worthy goal for quenchbody development.

There are numerous ways to potentially improve the antigen-dependent fluorescence fold-increase of quenchbodies without modification of tryptophan residues. For example, substituting fluorophore used here 5(6)-carboxytetramethylrhodamine (TAMRA) with similar dyes like ATTO520 or rhodamine 6 G can lead to improvements[28,44], although this is not a panacea and seems to vary depending on the quenchbody-antigen pair[18,28]. In addition, modification of the N-terminal amino-acid linker length where the fluorophore attaches, or changing the length of the C spacer in the fluorophore itself (e.g. C0–6), are other ways of potentially improving the quenchbody performance, provided it can improve the interaction of the fluorophore with quenching residues in the unbound state but not the antigen-bound state[11,18,28]. Furthermore, double-labelling quenchbodies with two fluorophores at positions that allow dye-dye H-dimer formation, causing additional dye quenching in the unbound state but not the antigen-bound state, can lead to further improvements, and are known as "ultra-quenchbodies"[38]. These ultra-quenchbodies exhibit fluorescence fold-increases as high as 50-fold in some cases, but this design can so far only be applied to scFv or Fab[38], with no nanobody versions currently existing. Finally, incorporation of an orthogonal quencher (e.g. quencher not based on natural amino acid side chains) into the CDRs, potentially via unnatural amino acid incorporation or bioconjugation, could potentially lead to quenchbodies with huge improvements in fold-sense. Ultimately, it would be interesting to investigate each of these as potential strategies for improving the current tryptophan-optimised quenchbody and will form the subject of ongoing investigations.

Considering IL-6 is present in picomolar quantities (5.2 pg/mL in the blood of healthy individuals[45]), the diagnostic usefulness of the IL-6 quenchbodies generated in this study (top $EC_{50}$ = 20 nM) is currently limited to scenarios where the IL-6 concentration is massively raised, such as septic conditions (1600 pg/mL[46]), or where IL-6 is pre-enriched. Affinity maturation of top-performing IL-6 quenchbodies through additional evolution and screening may potentially produce quenchbodies with sub-nanomolar affinity for IL-6 and therefore raise the efficacy of detection close to the requirements for diagnostic detection of IL-6 in blood plasma. In addition, a "double quenchbody" approach could be applied to increase the affinity, as it has been previously shown that dimeric ultra-high-affinity nanobodies can achieve binding affinities as low as ~0.03 nM[47]. Ultimately, work that aims to improve the sensitivity of quenchbodies is of great interest for generating a viable homogenous immunoassay which could eventually replace ELISA for the detection of low abundant analytes, such as IL-6. The ease of generation of quenchbodies, coupled with their entirely synthetic evolvability and rapid production within just a few weeks, supports the continued development of the quenchbody assay.

## Methods

### In silico modelling and molecular dynamics simulations of quenchbodies

Initial coordinates to inform quenchbody design were obtained from PDB ID: 1ZVH[32] and 5M14[30], corresponding to the X-ray structures of lysozyme and MBP-binding nanobodies, respectively. N-terminal Cys-tags were modelled using Modeller (version 10.2)[48,49]. Systems were set up using CHARMM-GUI[50]. The CHARMM36m protein force field was used for proteins. The CHARMM General Force Field (CGenFF) generated using the CGenFF interface at parachem.org (https://cgenff.umaryland.edu)[51] was used for the fluorophore, TAMRA. The TIP3P model was used for water[52]. Simulations were carried out using NAMD 3.0 (Nanoscale Molecular Dynamics, version 3 alpha)[53].

Simulations were performed after solvating the system in an octahedral box that extended at least 12 Å from the solute interface. $Na^+$ and $Cl^-$ counter ions were added to neutralise the system and achieve a salt concentration of 0.15 M. pKa calculations were performed using PROPKA[54] to assign protonation states of ionisable residues. Simulations were performed using periodic boundary conditions (PBC) at constant temperature (303.15 K) with the Langevin algorithm (a damping coefficient of 1/ps)[55] and at a pressure of 1.0 bar using the Nose-Hoover Langevin Piston method[56]. Hydrogen mass repartitioning[57] was applied with the time step set to 4.0 fs, and all covalent bonds involving hydrogens were kept rigid with the RATTLE algorithm[58]. Short-range electrostatics were calculated together with long-range electrostatics particle mesh Ewald (PME)[59] with a cut-off of 9.0 Å and a PME grid size of 1.0 Å. For all systems, energy minimisation (10,000 steps) and 125 ps equilibration were performed first with positional restraints placed on all the protein-heavy atoms (with a force constant of 1.0 kcal/mol/Å² on the backbone atoms and 0.5 kcal/mol/Å² on the side chain atoms) and TAMRA heavy atoms (with a force constant of 0.5 kcal/mol/Å²). This was followed by 12 and 6 μs production runs for the apo and antigen-bound quenchbodies, respectively. Six independent replicates for each system were simulated, with a total of 72 μs for the apo state and 36 μs for the antigen-bound states. Initial structures for the simulations were the top 3 modelled Cys-tag structures based on DOPE scores with attached TAMRA(R) or TAMRA(S). Snapshots were saved every 100 ps. VMD (Visual Molecular Dynamics)[60], LOOS (Lightweight Object-Orientated Structure Analysis)[61], MDAnalysis[62] and UCSF Chimera[63–65] were used to analyse and visualise the trajectories. Quenching distances were measured between the centres of masses of each of the tryptophans and TAMRA. FoldX was used to predict the effects of residue mutations on quenchbody stability and antigen binding[35].

### DNA and protein sequence design of quenchbodies

Protein coding sequences from either the MBP-binding nanobody or the lysozyme-binding nanobody were designed with (i) an N-terminal Cys-tag as a target for fluorophore labelling, (ii) a C-terminal Avi-tag to enable biotinylation, and (iii) a 3× FLAG-tag and 10× His-tag to facilitate purification procedures. The entire protein coding element was then converted to DNA and codon optimised using IDT's *Escherichia coli* (*E. coli*) B strain optimiser. This protein-coding DNA element was then combined into a gene expression cassette featuring a flanking T7 promoter and terminator for cell-free in vitro transcription and translation (IVTT) of the protein product and ordered as a Geneblock from IDT. The Geneblock also featured flanking DNA elements suitable for polymerase chain reaction (PCR) replication of the Geneblock, specifically DNA complementary to the forward primer 5′-ACCCGGCATGACAGGAG-3′ and the reverse primer 5′-TGGCGGCCGCTCTA-3′. PCR replication of Geneblocks was conducted using Q5® Hot Start High-Fidelity Master Mix (NEB; as per the manufacturer's instructions) at a scale of 100 μL, using 50 ng of Geneblock as template and 0.5 μM of forward and reverse primer in a PCR Mastercycler (Eppendorf), with initial heating at 98°C for 30 s, followed by 32 cycles of denaturation (98 °C, 10 s), annealing (60 °C, 30 s), and extension (72 °C, 15 s). Quenchbody PCR products were purified using Wizard Clean-up kits

(Promega; as per the manufacturer's instructions) and were quantified by $A_{260}/A_{280}$ absorbance on a Nanodrop 2000C spectrophotometer (ThermoFisher). Quenchbody PCR products were further analysed by agarose gel electrophoresis to ensure PCR products were of the expected size and purity before they were used for IVTT. All mutant quenchbodies were generated by in silico sequence modification of 5M14 and 1ZVH quenchbody constructs and re-ordered as Geneblocks from IDT.

### In vitro quenchbody expression, labelling and purification

For each quenchbody, 600 ng of purified PCR product was combined with NEBExpress® Cell-free *E. coli* Protein Synthesis System (NEB)[66] at 2× the reaction scale, including protein disulphide bond enhancer (NEB) and GamS (NEB), as per the manufacturer's instructions, and incubated for 16 h at room temperature with shaking at 1200 rpm in a Thermomixer (Eppendorf; used for all subsequent shaking and incubation steps). Expressed quenchbodies were then purified from the crude IVTT mixture (100 μL) by combining with 12.5 μL of Pierce Anti-DYKD4K (FLAG) Magnetic Agarose beads (equivalent to 50 μL of original resuspension) prewashed in phosphate-buffered saline (PBS) using a MagJET separation magnet (used for all subsequent wash steps). The crude IVTT-FLAG bead mixture was then mixed at room temperature for 30 min, 1200 rpm, to allow binding of the FLAG-tagged quenchbody to the beads, and subsequently washed with 0.2 mL PBS. In MBP-binding quenchbodies, the beads were subjected to an additional wash with 0.2 mL of 0.5 M maltose dissolved in PBS to remove endogenous MBP (present in the endogenous *E. coli* cell-free lysate as a contaminant), as maltose competes with the nanobody for MBP-binding[30]. The beads were then washed with 100 μL 1 mM tris(2-carboxyethyl)phosphine (TCEP) in PBS for 10 min at 16 °C with shaking at 1200 rpm, followed by washing with 0.2 mL degassed PBS. Beads were immediately combined with 100 μL 250 μM 5(6)-carboxytetramethylrhodamine (TAMRA) maleimide with C6-linker dissolved in degassed PBS at a final concentration of 1% (v/v) dimethylsulfoxide (DMSO), and incubated for 3 h (room temperature, 1200 rpm). Beads were then subjected to 9 × 0.2 mL washes with PBS to remove unconjugated dye. Finally, quenchbodies were eluted from the beads using 50 μL of 1.5 mg/mL Pierce™ 3× DYKDDDDK Peptide (ThermoFisher).

### SDS-PAGE and in-gel fluorescence analysis

Proteins were diluted in 2× denaturing Tris-glycine sample buffer to a final concentration of 63 mM Tris-HCl, 10% (v/v) glycerol, 5% (v/v) β-mercaptoethanol, 2% (w/v) sodium dodecysulfate (SDS), 0.0025% (w/v) bromophenol blue, pH 6.8, and subjected to SDS-polyacrylamide gel electrophoresis (SDS-PAGE) at 150 V for 30 min using Mini-PROTEAN TGX precast gels in a Tetra electrophoresis system filled with Tris-glycine running buffer (25 mM Tris, 192 mM glycine, 0.1% (v/v) SDS, pH 8.3), according to the manufacturer's instructions (BIO-RAD, Gladesville, Australia). Protein size was estimated using Precision Plus Dual Colour Protein Standards (BIO-RAD). All gels were stained overnight with Instant Blue (Expedeon, Cambridge, UK) and subsequently destained in MilliQ $H_2O$ overnight before being imaged on a Gel Doc XR+ Molecular Imager (BIO-RAD). In-gel TAMRA fluorescence of quenchbodies following SDS-PAGE was analysed on an Amersham Typhoon Biomolecular Imager (Cytiva), using the preset acquisition settings for Cy3 imaging (excitation = 532 nm, emission filter = 570 ± 20 nm).

### Pulldown binding assays

To assess their antigen binding, quenchbodies were expressed as above and were purified from the crude IVTT mixture (100 μL) by combining with 12.5 μL beads/50 μL suspension of Pierce Anti-DYKD4K (FLAG) Magnetic Agarose (30 min, 1200 rpm). The crude IVTT-FLAG bead mixture was then mixed at room temperature for 30 min, 1200 rpm, to allow binding of the FLAG-tagged quenchbody to the beads, and subsequently combined with 100 μL of (i) 1 μM MBP for MBP-binding quenchbodies, or (ii) 10 μM lysozyme for lysozyme-binding quenchbodies, and incubated for 30 min (room temperature, 1200 rpm). Beads were then subjected to 6 × 0.2 mL

washes with PBS to remove any unconjugated or non-specifically bound antigen, and finally eluted with 50 μL of 1.5 mg/mL Pierce™ 3× DYKDDDDK Peptide. The eluted quenchbody-antigen complex obtained by FLAG-pulldown was subjected to reducing SDS-PAGE, and the effect of mutations on antigen binding was semi-quantitatively analysed by comparison to the binding of the wild-type (WT) control.

### Protein:TAMRA quantification

Protein concentration of TAMRA-labelled quenchbodies was quantified by $A_{280}$ absorbance on a Nanodrop 2000C spectrophotometer and was corrected against a FLAG peptide standard in PBS, subjected to the same elution procedures to correct for background absorbance associated with the FLAG peptide. Protein concentration was further calculated by including the correction factor for TAMRA ($A_{280} = 0.178$) and assuming an extinction coefficient based on the polypeptide sequence properties calculated in ExPaSy ProtParam[67]; (approximately $\varepsilon = 47{,}000$ cm$^{-1}$ M$^{-1}$ for quenchbodies). TAMRA concentration of TAMRA-labelled quenchbodies was determined by further $A_{555}$ quantification, assuming an extinction coefficient of 90,000 cm$^{-1}$ M$^{-1}$ for TAMRA.

### Fluorescence spectrophotometry plate assays

Quenchbodies were diluted to 20 nM in PBS/0.05% (v/v) Tween-20 (PBST) in the absence or presence of 8000, 4000, 2000, 1000, 500, 250, 125, 64, 32, 16, 8, 4, 2, or 1 nM cognate antigens (lysozyme, MBP, or IL-6), or 2% (w/v) SDS with 5% (v/v) β-mercaptoethanol as denaturant, and incubated for 1 h at room temperature. Samples were subsequently dispensed ($n = 3$, 40 μL/well) into black 384-well Griener microplates and analysed by fluorescence spectrophotometry in a CLARIOstar microplate reader with excitation = 535 ± 20 nm and emission = 585 ± 30 nm, using a dichroic filter of 557.5 nm. To compare fluorescence increases that occur in the presence of various concentrations of antigen ("fold-sense"), all raw fluorescence values were normalised as a ratio to the 0 nM quenchbody control lacking any antigen or denaturant. To derive a quenchbody binding affinity, the fluorescence signal of quenchbody with 0 nM antigen was subtracted from all other samples containing antigen and the data were fitted to an equation describing a single site-specific binding mode with Hill slope to derive an equilibrium dissociation constant ($K_D$) in GraphPad Prism 9.4.0. The WT Qb-MBP was expressed and tested on three separate occasions, including temporal separation between replicates, and showed essentially identical results across replicates, indicating biological replicates were redundant when studying any of the other mutants in the study.

### Quenchbody library assembly

Quenchbody libraries with evolvable CDRs were assembled using oligonucleotides randomised by trimer phosphoramidite mix (TRIM) as previously described[30], with the following modifications. CDRs 1–3 were assembled by PCR assembly using Vent polymerase (NEB), with each reaction consisting of 1× ThermoPol buffer, 5% (v/v) DMSO, dNTPs (0.4 mM), forward and reverse outer primers (1 μM), random TRIM oligonucleotide (50 nM), megaprimer (25 nM), assembly primer (25 nM) and Vent polymerase (2 U), with a final volume to 100 μL. Megaprimers were purchased as 4 nmol ultramers (IDT). Random TRIM oligonucleotides were purchased either from IDT or Ella Biotech GmbH. The CDR 1 reaction consisted of CDR1_c random TRIM oligonucleotide, megaprimer1_c, FW2_C_rev assembly and FW1_c_for and Link1_c_rev outer primers. CDR2 reactions contained Q_CDR2_c randomised oligonucleotide, megaprimer2_c and Link1_c_for and Link2_c_rev outer primers. CDR3 reactions consisted of Q_CDR3_c randomised oligonucleotide, and Link2_c_for and FW4_c_rev outer primers. PCR assembly reactions were: 2 min initial denature at 94 °C, then 30 cycles of 94 °C 30 s, 60 °C 30 s, 72 °C 30 s, then final extension for 5 min at 72 °C. Individual reactions were purified on Macherey-Nagel Gel clean-up columns. DNA fragments encoding CDRs 1 and 2 were digested overnight with BsaI-HFv2 and BbsI-HFv2, respectively (NEB). Reactions were cleaned up with Macherey-Nagel

Gel cleanup columns. DNA products were quantified and ligated in equimolar mixtures. The correct ligation product was gel extracted before PCR amplification with outer oligos FW1_c_for and Link2_c_rev. CDR1 + 2 fragment was digested with BsaI-HFv2 and CDR3 fragment with BbsI-HFv2 and ligated with T4 ligase. The correct ligation product was gel extracted. BbsI restriction sites were added to the ends of the assembled library with 8 cycles of PCR. PCR products were then digested with BbsI and ligated to an expression vector using T4 ligase.

## SNAP display selection

Genes coding for nanobody variants were fused to the N-terminus of a SNAP-tag via a Glycine/Serine linker sequence, as part of an expression cassette under the T7 promoter. Genes were fused to dendrimer-like DNA structures (DL-DNA), such that each gene displayed eight benzylguanine moieties, as described previously[36,37]. To assemble nanobody library expression cassettes, we ligated 50 ng of *Bbs*I-digested nanobody library to 200 ng of *Bbs*I-digested expression vector with complementary overhangs for 1 h at 16 °C. Ligation products were then amplified with "outPCR" primers, containing uracil and overhangs compatible with Thermolabile USER II digestion/ligation (NEB). Following outPCR, DL-DNA were assembled by simultaneous Thermolabile USER II digestion and T4 ligation to DL-DNA structures with compatible cohesive ends. Ligation products were confirmed by agarose gel electrophoresis. The genes were diluted to 125 pM in 30 µL PUREExpress cell-free expression mix (NEB) containing 0.06% pluronic acid (F-127, final concentration) and 1 µL of each disulfide bond enhancer 1 and 2 (NEB #E6820). We used droplet oil consisting of 3 M™ Novec™ HFE7500 Engineered Fluid (3 M, 7100025016), filtered with a 2 µm cellulose filter (VWR, #514-0061) and supplemented with 2% fluorinated surfactant FluoSurf neat (Emulseo, #1903) for droplet emulsification. For emulsification, 0.4 mL of droplet oil was loaded into a 1 mL Luer Lock syringe, and the cell-free expression mixture layered on top. A 1 mL syringe filled with 0.8 mL of droplet oil connected to a 5 µm hydrophobic membrane pumping device supplied by Shirasu Porous Glass (SPG) (#PC05U). The oil phase was pushed through the SPG membrane until the oil phase was level with the open side. The syringe containing the cell-free expression mix was then connected to the SPG membrane, and the mixture was emulsified by repeatedly alternating the mixture from one syringe to the next, for a total of 10 rapid extrusions through the membrane. The emulsification process produces monodisperse droplets with volumes in the fL range. At ≈125 pM in PUREExpress cell-free expression mix, droplet loading is 0.1 genes per droplet. The emulsion was then transferred into a new 1.5 mL tube and incubated at 37 °C for 4 h, then at 4 °C overnight in an Eppendorf thermomixer. To break the emulsion, excess HFE oil was removed by aspirating with a 23 G needle from the bottom of the tube, and Perfluoro-1-Octanol (Sigma-Aldrich/Merck, #370533) was added to the tube in 1:1 ratio with the emulsion volume. After vortexing for 15 s, the tubes were centrifuged at 13,000 rpm for 1 min, and the aqueous phase was recovered in a clean tube. The recovered emulsion was diluted in 5 x Recovery and Binding Buffer (RBB; 40 mM Tris-HCl, pH 7.4, 10 mM EDTA, 1 mM dithiothreitol, 0.5% Tween-20, and 10 µM BG-mPEG12). Magnetic beads displaying recombinant IL-6 protein (BioLegend #570806) were prepared by washing 70 µL of tosyl-activated M-280 magnetic dyna-beads (ThermoFisher #14203), 3× with 1 mL of PBS for 5 min. IL-6 binding buffer was prepared by diluting 10 µg of IL-6 in PBS to a total volume of 75 µL. Beads were coupled to 70 µL of IL-6 binding buffer by incubating overnight with end-over-end mixing at 37 °C. The quantity of IL-6 bound to beads was determined by absorbance spectroscopy of pre- and post-binding solutions on a nanodrop at A280. Following binding, beads were washed 1× with 1 mL PBS, then with 1 mL 50 mM Tris-HCl (pH 7.5), 0.5% Tween-20 for 1 h with end-over-end mixing. Finally, beads were washed for 1 h in 1 mL of PBS for 1 h before being washed 1× and resuspended in 20 µL 50 mM Tris-HCl (pH 7.5), 0.5% Tween-20. SNAP-display complexes were then selected by affinity panning on IL-6-coated beads (500 nM equivalent IL-6, unless otherwise stated), in a total volume of 75 µL (2 µL of solution was retained for qPCR analysis) for 1 h at room temperature with end-over-

end mixing. Selections were washed 5× in 200 µL of RBB. Bound complexes were then released from the beads by heat shock (20 min at 70 °C) in 15 µL of 50 mM Tris-HCl (pH 7.5), 0.5% Tween-20, and the selected genes amplified by PCR using Q5® Hot Start polymerase (NEB) using forward primer: GATCACGAAGACATTATGGCGGATCAAGTCCAGCTGGT GGAATCG and reverse primer: GATCACGAAGACATCACCAGAAC GGTAACTTGGGTGCCCTG. PCR reactions were composed of 10 µL Q5 reaction buffer (NEB), 1 µL 10 mM dNTPs (NEB), 2 µL each 10 µM forward and reverse primers, 5 µL elution, 24.5 µL nuclease-free water, 5 µL DMSO, and 0.5 µL Hot-start Q5 high fidelity polymerase. Cycling conditions were 98 °C, 2 min, followed by 25–30 cycles of 98 °C 30 s, 60 °C 20 s and 72 °C 30 s, with a final extension of 72 °C for 2 min. Amplification of recovered genes was verified by agarose gel electrophoresis. Selected genes can then be re-formatted for further rounds of selection.

## Statistics and reproducibility

Statistical analyses were done using GraphPad Prism 9.4.0. Binding curves were fitted to a single site-specific binding model with a Hill slope to derive equilibrium dissociation constants. Fluorescence spectrophotometry plate assays were carried out with three technical replicates per condition, and data are reported as mean ± standard deviation (SD). For the MBP-targeting quenchbody (Qb-MBP), experiments were independently repeated on three separate occasions with temporal separation, yielding essentially identical results. Molecular dynamics simulations were conducted as six independent replicates per system, with a total of 72 µs for the apo and 36 µs for the antigen-bound states. For in vitro evolution experiments, two technical replicates of NGS selections were performed, and enrichment was assessed across five selection rounds. Where statistical significance was tested (e.g. IL-6 titration), one-way ANOVA with Tukey's multiple comparisons was used, and significance was defined at $\alpha = 0.05$.

## Reporting summary

Further information on research design is available in the Nature Portfolio Reporting Summary linked to this article.

## Data availability

Data supporting the findings of this study are available on a Zenodo repository (https://doi.org/10.5281/zenodo.14921109).

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

## Acknowledgements

The authors would like to thank members of the Spenkelink, van Oijen, and Griffiths labs, and Quantum-Si for helpful discussions. This work was supported by the Australian National Health and Medical Research Council (Investigator grant 2007778 to L.M.S.), the Human Frontiers of Science Programme (Programme grant RGP0021/2020 to A.D.G. and A.M.v.o.), the Australian Research Council Centre of Excellence in Quantum Biotechnology (CE230100021) and with the assistance of resources and services from the Australian National Computational Infrastructure (NCI).

## Author contributions

Conceptualisation: J.H.C., N.S.E., S.H., C.M.G., R.R.S., N.S., V.Z., H.G., M.R.C., H.Y., A.M.v.O., A.D.G. and L.M.S.; Methodology: J.H.C., N.S.E., S.H., C.M.G., S.H.M., R.R.S., N.S., H.G., M.R.C. and H.Y.; Software: N.S.E., S.H., C.M.G., S.H.M. and H.Y.; Validation: J.H.C., N.S.E., S.H., C.M.G., S.H.M., R.R.S., NS, HG, MRC, HY; Formal analysis: JHC, NSE, GHM, SH, CMG, RRS, APK, V.Z. and M.R.C.; Investigation: J.H.C., N.S.E., G.H.M., S.H., C.M.G., R.R.S., A.P.K., V.Z., H.G., M.R.C. and H.Y.; Recourses: J.H.C., G.H.M., S.H., C.M.G., S.H.M., A.P.K., V.Z. and H.Y.; Writing: J.H.C., N.S.E., S.H., C.M.G., H.Y., A.M.v.O., A.D.G. and L.M.S.; Visualisation: J.H.C., N.S.E., S.H., C.M.G. and L.M.S.; Supervision: H.G., M.R.C., H.Y., A.M.v.O., A.D.G. and L.M.S.; Funding acquisition: H.Y., A.M.v.O., A.D.G. and L.M.S.

## Competing interests

This work was financially supported by Quantum-Si. Marco Ribezzi-Crivellari and Sebastian Hutchinson are employed by Quantum-Si. Antoine van Oijen and Andrew Griffiths were members of the Quantum-Si Scientific Advisory Committee.
