## [Peer review file · Communications Biology]

Optimised Nanobody-based Quenchbodies for Enhanced Protein Detection

Corresponding Author: Dr Lisanne Spenkelink

This manuscript has previously been submitted to another journal. This document only contains information relating to versions considered at Communications Biology.

Version 0:

Reviewer comments:

Reviewer #1

(Remarks to the Author)

The manuscript presents detailed engineering of a nanobody to generate a universal Quenchbody scaffold that is amenable to further specificity engineering using *in vitro* selection. The approach and methodology are sound, and clear increases in signal to noise over previously described iterations are demonstrated for protein antigens. My main concern is the lack of specificity testing for the Quenchbodies described. Hence in Figures 2 and 4 the MBP, lysozyme, and IL-6 Q-bodies should be tested against at least one other protein antigen to demonstrate specificity. For the IL-6 Q-bodies, I appreciate that the *in vitro* selection approach inherently applies pressure on specificity, but formal demonstration is still required.

Minor comments

- 1) Line 186: correct "X-fold"
- 2) Figure S8: Perhaps indicate Selection 3 is blank beads control in Figure for easier comprehension.
- 3) Have the authors tested their novel Q-body scaffold against other classes of antigens to see if fold-sense can be further improved for these?

Reviewer #2

(Remarks to the Author)

This is a well-executed and well-written manuscript describing evolution of quenchbodies – nanobodies with conjugated fluorophores (at N-termini) quenched in their unbound states by tryptophans placed at specific locations. The authors coupled a computational study with laboratory evolution and testing of individual constructs to show that when the location of the tryptophan(s) residue(s) is optimized, the turn-on in fluorescence due to unquenching of the TAMRA fluorophore can increase, improving the detection of the target protein. Overall, this study is timely and points to an important advancement in the field of bioanalytical chemistry. I only have a few minor questions and edits/suggestions that the authors should address:

1. Why was TAMRA chosen as the fluorophore of choice and why was the linker relatively long? The Marmé et al (Bioconjugate Chem 2003 <https://doi.org/10.1021/bc0341324>) paper would suggest that BODIPY-type dyes would be much better quenched by tryptophans. The dye choice should be discussed a bit more in the manuscript.
2. Given that the nanobody starts with an N-terminal Cys residue, have the authors considered conjugating it with cyano-benzothiazole? This is a published biorthogonal chemistry that would yield N-terminal luciferin, which is a decent fluorophore. Importantly, the dye is formed by Cys so unreacted benzothiazole does not give any signal.
3. Would inclusion of solute quenchers, such as acrylic acid or iodide, help some of these constructs? I imagine that further differential quenching could be achieved to get higher turn on. Clearly in cases where the fluorophore goes from touching a Trp to essentially free-in-solution, this would not benefit from soluble quenchers, but perhaps some of the constructs that showed no change might benefit from them.
4. Another discussion point, perhaps: it seems that combining unquenching with polarization would increase the signal further, especially when binding a large target protein.

5. Figures:

- a. All figures would benefit from having the N and C termini labeled on the proteins. This is particularly useful because the N-terminus is where the dye is conjugated.
- b. Fig. 3: Please label the Trp residues like in other figures
- c. Fig S2, panel D: the pulldown experiment is mislabeled as MBP, but should be Lys
- d. Fig S5: please move the arrows to point to the bands

Andrej Luptak

Reviewer #3

(Remarks to the Author)

In this manuscript, the authors optimized the antigen-dependent change in fluorescence quenching of N-terminal fluorescently labeled nanobodies. They identified the positions of Trp residues effective for quenching in two existing nanobodies, and obtained quenchbodies for IL6 from a random library containing these Trp residues. They demonstrated that the obtained clones and their affinity maturation mutants showed relatively good binding and fluorescence response. However, the fluorescence change was not significantly improved, which indicates the limitations of the authors' strategy. Although the limitations may be overcome by, for example, further analysis of the obtained clones (#1-#36), I consider that the manuscript at this point does not have significant advances worthy of being accepted by Communications Biology. In addition, I consider the following points should be improved.

- The binding activity of Trp-to-Tyr mutants (Fig. 3) should be compared not only by qualitative pull-down but also by quantitative EC50. More consideration should be given to the effect of Trp substitution in the CDR on antigen binding.
- There is no data on single Trp substitution in MBP, and there is no explanation for Fig. 3E. Therefore, there is no basis to assume that W110 and W115 are effective in quenching and that the nearby Y114W can also participate in quenching. Is W-to-A mutants in Fig. 3E a mistake for W-to-Y mutants?
- The data for Lysozyme and MBP were confused. Lysozyme Y-to-W mutants are Fig. 3C, not Fig. 3B. MBP Y-to-W mutants are Fig. 3F, not Fig. 3C.
- The effectiveness of the Trp residue in the nanobodies against IL6 has not been verified. For example, the fluorescence response of nanobodies with Trp residue substitutions in #15 should be examined.

Version 1:

Reviewer comments:

Reviewer #1

(Remarks to the Author)

The authors have addressed my concerns.

Reviewer #2

(Remarks to the Author)

Looks good. Publish.

Minor typo: Line 385 "sec" should be "second"

Reviewer #3

(Remarks to the Author)

I appreciate the authors' sincere response. However, compared to the original scFv Quenchbody (one of which showed a maximum fluorescence change of 5.6-fold (ref. 11)), the increase from 1.3-1.5-fold to 1.5-2.4-fold is an insignificant change and is insufficient to prove the usefulness of the authors' strategy, even considering the advantages of nanobody.

I confirmed the correction in the explanation of Fig. 3E. However, I do not agree with the conclusion that all three Trp residues are involved in quenching based on the result that the W-to-A triple mutant of anti-MBP showed no response, without examining the binding activity by pull-down. This result cannot exclude the possibility that the W-to-A triple mutant has lost binding activity and therefore the Trp residues involved in quenching cannot be identified, and the possibility that only one or two of the three Trp residues are involved in quenching.

I appreciate that the authors conducted additional experiments with W-to-Y mutants of Variants #1, #15, and #36 in accordance with my comments. However, the authors carried out the similar inappropriate experimental plan. The result that the W-to-Y quintuple mutant showed no response does not exclude the possibility that the binding activity was lost and therefore the quenching effect of Trp residues cannot be examined, and the possibility that only some Trp residues are involved in quenching and other Trp residues are worthless.

I agree the usefulness of Quenchbody libraries containing Trp residues that may be involved in quenching, and I commend the authors for investigating the usefulness. However, I consider this manuscript insufficient to be accepted for three

reasons: the result in Fig. 3E does not provide evidence that W101, W110, and W115 should be included in the library; the effects of individual Trp in the anti-IL6 variants have not been examined; and the improvement in response is not significant.

Response to reviewers

We are grateful to the reviewers for their thoughtful and constructive feedback, which has helped us improve the quality of our manuscript. We have carefully considered each of the reviewers' comments, carried out additional experiments, and revised the text, as detailed in our point-by-point response below.

Reviewer #1:

The manuscript presents detailed engineering of a nanobody to generate a universal Quenchbody scaffold that is amenable to further specificity engineering using in vitro selection. The approach and methodology are sound, and clear increases in signal to noise over previously described iterations are demonstrated for protein antigens. My main concern is the lack of specificity testing for the Quenchbodies described. Hence in Figures 2 and 4 the MBP, lysozyme, and IL-6 Q-bodies should be tested against at least one other protein antigen to demonstrate specificity. For the IL-6 Q-bodies, I appreciate that the in vitro selection approach inherently applies pressure on specificity, but formal demonstration is still required.

We appreciate this point about demonstrating specificity and agree it is important to show that the quenchbodies do not react non-specifically to the presence of other protein antigens. We have since carried out additional experiments and have now included the following statement in the results section which describes this work: "We chose variants 15, 33 and 36 for specificity testing, which showed similar, only slightly reduced fluorescence increases in the presence of IL-6 after spiking with 50% human plasma (Fig S15), which may reflect non-specific quenching of TAMRA by high concentrations of plasma proteins" and have provided corresponding data as per Fig S15. We believe that spiking with a complex protein mixture of proteins as found in human plasma, and the resulting absence of increased fluorescence intensity of the IL-6 quenchbodies even in 0 nM IL-6, demonstrates a lack of reactivity to non-specific proteins.

Minor comments

1) Line 186: correct "X-fold"

This has been changed to reflect the correct value of 1.6–2-fold.

2) Figure S8: Perhaps indicate Selection 3 is blank beads control in Figure for easier comprehension.

In line with the recommendations of reviewer #1, we have changed "Selection 3" to "Blank beads control" in Fig S8 to aid comprehension.

3) Have the authors tested their novel Q-body scaffold against other classes of antigens to see if fold-sense can be further improved for these?

We refer to this statement in the introduction "Nanobody-based quenchbodies are particularly effective for recognition of smaller antigens like small-molecule drugs or peptides [25], with a maximal 6-fold increase in fluorescence intensity observed upon binding of a

quenchbody to methotrexate [17]. However, recognition of larger protein antigens generally provides poorer responses [25], with fluorescent fold-increases of only 1.1–1.7 observed for albumin [11, 18], claudin [26], human epidermal growth factor receptor-2 [27], and hemagglutinin [28].” Thus, we limited the scope of the present study to improve the response of quenchbodies against only proteins, but we expect that the present scaffold could be used to obtain quenchbodies with higher fold-sense against other antigens in future work. We have thus added an additional statement to the discussion “In addition, the present scaffold may hold superior detection performance for non-protein based antigens, such as small molecules, metabolites or other biomolecules which lack quenching amino acids.”

Reviewer #2:

This is a well-executed and well-written manuscript describing evolution of quenchbodies – nanobodies with conjugated fluorophores (at N-termini) quenched in their unbound states by tryptophans placed at specific locations. The authors coupled a computational study with laboratory evolution and testing of individual constructs to show that when the location of the tryptophan(s) residue(s) is optimized, the turn-on in fluorescence due to unquenching of the TAMRA fluorophore can increase, improving the detection of the target protein. Overall, this study is timely and points to an important advancement in the field of bioanalytical chemistry. I only have a few minor questions and edits/suggestions that the authors should address:

We thank the reviewer for their positive assessment of our work, and for their thoughtful suggestions.

1. Why was TAMRA chosen as the fluorophore of choice and why was the linker relatively long? The Marmé et al (Bioconjugate Chem) paper would suggest that BODIPY-type dyes would be much better quenched by tryptophans. The dye choice should be discussed a bit more in the manuscript.

We thank the reviewer for this insightful question. We were limited to the use of a 532 nm-excitable fluorophore by Quantum-SI who funded the research for obtaining sensors which would be compatible with their Platinum™ device which exclusively uses this laser. We chose TAMRA as the fluorophore and used a Cys-tag with no linker as the conjugation site (which we note is relatively long), as this was identified in previous research which screened multiple fluorophores and linker lengths, and found this to be the combination with the highest performance. We have correspondingly added this statement to the introduction “We chose TAMRA conjugated via an N-terminal Cys-tag on the quenchbody with no linker, as this was identified to give the highest response in previous research which screened multiple linker lengths and fluorophores for nanobody-based quenchbodies [18].”. Correspondingly, we did not consider using BODIPY given its lower excitation near 502 nm, but appreciate that the testing of additional fluorophores is an interesting line of query for future research that could lead to further improvements and are thankful for this insightful query.

2. Given that the nanobody starts with an N-terminal Cys residue, have the authors considered conjugating it with cyano-benzothiazole? This is a published biorthogonal chemistry that would yield N-terminal luciferin, which is a decent fluorophore. Importantly, the dye is formed by Cys so unreacted benzothiazole does not give any signal.

Given we have now added more information regarding our design choices as provided in answer to 1., which now forms part of the manuscript's introduction, we expect that this will help answer why we did not consider this strategy. Specifically, luciferin has an excitation maximum around 385 nm, which was outside our design constraint which necessitated the use of 532-excitable fluorophores. However, this is a very interesting and insightful query that we will investigate in future work, and we thank the reviewer for their feedback.

3. Would inclusion of solute quenchers, such as acrylic acid or iodide, help some of these constructs? I imagine that further differential quenching could be achieved to get higher turn on. Clearly in cases where the fluorophore goes from touching a Trp to essentially free-in-solution, this would not benefit from soluble quenchers, but perhaps some of the constructs that showed no change might benefit from them

This is an interesting question and we have carried out additional experiments to see if a null-responding quenchbody could be improved by the addition of an in-solution quencher (100 mM potassium iodide), and compared it to the response of a low-responding quenchbody, as recommended. We have added the following statement to the results section which describes the outcome of these experiments: "Addition of an in-solution quencher (100 mM potassium iodide) did not change the fluorescence response of null-responder (variant #27) and slightly reduced the response of a low-responder (variant #15) (Fig. S13), suggesting the fluorophore in the antigen-bound state is predominantly exposed to solvent." We believe these additional experiments will address the question of the reviewer and strengthens the manuscript by providing additional evidence for the translocation of the fluorophore during antigen-binding.

4. Another discussion point, perhaps: it seems that combining unquenching with polarization would increase the signal further, especially when binding a large target protein.

Given we have now added more information regarding our design choices as provided in answer to 1., which now forms part of the manuscript's introduction, we expect that this will help answer why we did not consider this strategy. Specifically, that we were primarily interested in use of quenchbodies which could be measured by changes in fluorescent intensity, for potential use on the Platinum™ device. In addition, fluorescence polarisation of quenchbodies has been previously examined in detail by another research group which found that magnitude of increase of fluorescence polarisation ($1/r$) was generally much smaller than the magnitude of increase observed by fluorescent intensity (DOI:10.1021/acs.bioconjchem.6b00217, see Figure 3 in this ref). We have also now added this information and the associated reference to the introduction in the following statement "This flexible linker allows the fluorophore to interact with tryptophan residues in the quenchbody. This interaction leads to a small change in fluorescence polarisation ($1/r$) but a more marked reduction in fluorescence intensity (quenching) through photo-induced electron transfer (Fig. 1A) [14].

5. Figures:

a. All figures would benefit from having the N and C termini labeled on the proteins. This is particularly useful because the N-terminus is where the dye is conjugated.

We have added N and C termini labels on the proteins to all relevant figures to aid comprehension.

b. Fig. 3: Please label the Trp residues like in other figures

We have added Trp (W) residue labels to figure 3 and 4 to aid comprehension.

c. Fig S2, panel D: the pulldown experiment is mislabeled as MBP, but should be Lys

We have changed the mislabelling of MBP to Lys in Fig S2, as this was a typo.

d. Fig S5: please move the arrows to point to the bands”

We have moved the arrows in Fig. S5 to point more accurately to the bands, as they were slightly misaligned.

Reviewer #3:

In this manuscript, the authors optimized the antigen-dependent change in fluorescence quenching of N-terminal fluorescently labeled nanobodies. They identified the positions of Trp residues effective for quenching in two existing nanobodies, and obtained quenchbodies for IL6 from a random library containing these Trp residues. They demonstrated that the obtained clones and their affinity maturation mutants showed relatively good binding and fluorescence response.

However, the fluorescence change was not significantly improved, which indicates the limitations of the authors' strategy. Although the limitations may be overcome by, for example, further analysis of the obtained clones (#1-#36), I consider that the manuscript at this point does not have significant advances worthy of being accepted by Communications Biology.

We thank reviewer #3 for their feedback. Respectfully, we disagree with their comment on the improvement of the quenchbody performance and believe this may be a misunderstanding on behalf of reviewer #3. We refer to this statement in the discussion section which summarises how we significantly improved the quenchbody scaffold, which had a fluorescence response of 1.3–1.5, and increased to 1.5–2.4 as an end result of our strategy. “Using this scaffold in an *in vitro* evolution screen, we were able to generate novel quenchbodies for the detection of IL-6 with 1.5–2.4 fold-sense. This is a marked improvement over the fold-sense of 1.3–1.5 obtained for the quenchbodies we generated as models against MBP and lysozyme in the present study, and fluorescence fold-increases of 1.1–1.7 reported for other nanobody-based quenchbodies against proteins [11, 19], [27-29].”

We also refer to the comment of Reviewer #1 “The approach and methodology are sound, and clear increases in signal to noise over previously described iterations are demonstrated for protein antigens.” and the comment of reviewer #2 “The authors coupled a computational

study with laboratory evolution and testing of individual constructs to show that when the location of the tryptophan(s) residue(s) is optimized, the turn-on in fluorescence due to unquenching of the TAMRA fluorophore can increase, improving the detection of the target protein.”, which further supports the success of the strategy in improving the fluorescence change for target proteins. We hope that this information helps clarify any potential misunderstanding of reviewer #3.

In addition, I consider the following points should be improved.

- The binding activity of Trp-to-Tyr mutants (Fig. 3) should be compared not only by qualitative pull-down but also by quantitative EC50. More consideration should be given to the effect of Trp substitution in the CDR on antigen binding. “The binding activity of Trp-to-Tyr mutants (Fig. 3) should be compared not only by qualitative pull-down but also by quantitative EC50. More consideration should be given to the effect of Trp substitution in the CDR on antigen binding.”

We thank the reviewer for this thoughtful recommendation, and this is something we have considered. However, determination of EC50 after mutation of tryptophan residues only provides quantitative information about the strength of binding, but does not report on performance as a fluorescent sensor, which is a combination of binding affinity, quenching, and dequenching. (*E.g.* quenchbodies that bind with poor affinity (high EC50) can show high fluorescence response, and *vice versa*). This is evident from the library where different fluorescence responses of quenchbodies are seen for different EC50 values (Fig. S11D). Hence, we carried out qualitative pull-downs to confirm presence of antigen, to determine if mutations made to increase the binding performance could be considered relevant (depending on whether the antigen was present). In general, the amount of antigen pulldown observed did not correlate with fluorescence performance. For example, mutant Y54W/Y114W which showed similar pulldown to Y33W/Y59W/Y114W and Y54W/Y59W/Y114W (Fig S6.) had worse fluorescence response (Fig 3F). Therefore, we believe determination of EC50 does not provide information that will add value and strengthen the manuscript.

- There is no data on single Trp substitution in MBP and there is no explanation for Fig. 3E. Therefore, there is no basis to assume that W110 and W115 are effective in quenching and that the nearby Y114W can also participate in quenching. Is W-to-A mutants in Fig. 3E a mistake for W-to-Y mutants?

We thank the reviewer for pointing this out and confirm that there were some typographic errors which incorrectly referred to some panels of Figure 3 out of order when described in the text, namely a missing description for Fig. 3E. We have corrected this typo which can be found in the following passage, which we believe will now answer the query of reviewer #3. Specifically, description for mutation of W101A, W110A and W115A is now provided which confirms they are effective in quenching (Figure 3E), and this is not a mistake for W-to-Y mutants.

“First, we substituted intrinsic CDR tryptophans which contact the antigen (Fig. 3D) for alanines to confirm the tryptophans most important for quenching. Knockout with W101A,

W110A and W115A showed no response supporting the importance of these tryptophans (Fig. 3E). We next substituted orthogonal CDR-tyrosines that contact the antigen with tryptophans, aiming to improve the quenchbody performance. A Y114W single mutant was first produced as it was considered potentially redundant based on its proximity to existing W110 and W115. The Y114W mutant retained its binding affinity and slightly improved the quenchbody fold-sense to 1.6 (the WT is 1.5). Therefore, the Y114W mutant was kept constant in further single, pairwise and triple mutations. The Y59W/Y114W mutant had an improved fold-sense of 1.9, suggesting a large improvement due to Y59W (Fig. 3 F). For all other mutants fold-sense was lower than the Y114W mutant, and in some cases lower than WT (Fig. 3 C), which was possibly a result of complete loss of binding to the antigen as observed by pulldown assay (Fig. S6).”

- The effectiveness of the Trp residue in the nanobodies against IL6 has not been verified. For example, the fluorescence response of nanobodies with Trp residue substitutions in #15 should be examined.

We agree that the effectiveness of the nanobodies against IL-6 should be verified by tryptophan substitution. We therefore carried out additional experiments to determine the fluorescence response of W-to-Y substitution in variant #15 as recommended by reviewer #3, and also compared this to similar experiments carried out on variants #13 and #36 to deepen the characterisation across the library. We have provided additional data to the manuscript, and the outcome of these experiments is now described in the following results statement and in Fig. S14. “Substitution of favourable tryptophans with tyrosines, to produce W59Y/W101Y/W103Y/W110Y/W115Y combinatorial mutants showed a complete lack of fluorescence response to IL-6 in variants #13, #15 and #36 (Fig. S14), supporting the importance of these tryptophan positions for quenchbody performance.” We believe that this data provides additional evidence of the importance of these tryptophan positions and helps to strengthen the manuscript further.

We thank the editor for the opportunity to revise our paper to address the remaining concerns of the reviewers. We have added text to the Results and Discussion sections highlighting the limitations of our study, as commented by reviewer #3. Below is our point-by-point response to the reviewer's comments.

Reviewer #3 (Remarks to the Author):

I appreciate the authors' sincere response. However, compared to the original scFv Quenchbody (one of which showed a maximum fluorescence change of 5.6-fold (ref. 11)), the increase from 1.3-1.5-fold to 1.5-2.4-fold is an insignificant change and is insufficient to prove the usefulness of the authors' strategy, even considering the advantages of nanobody. The scFV-based quenchbody highlighted in ref. 11 recognises a small peptide (< 10 amino acids). As highlighted in paragraph 4 of the introduction, short peptides are an easier class of antigen to obtain quenchbodies with higher fluorescent increases, likely due to the absence of quenching residues in the target peptide. In contrast, our Quenchbodies detect entire proteins. To our knowledge, the highest-reported fold sense for purified nanobody-based quenchbodies detecting entire proteins is 1.4 (ref. 19), which is markedly lower than the response of the quenchbodies obtained using our strategy (1.5-2.4-fold).

Furthermore, the fluorescence responses of nanobody-based quenchbodies cannot be compared directly with those of scFv-based quenchbodies in reference 11. All antibody molecules have markedly different biochemical considerations, including their own advantages and disadvantages. The advantages of nanobody as noted by the reviewer and in this manuscript are indeed why any larger fluorescence improvements observed in nanobody-based quenchbodies are more impactful, than say larger fluorescence fold increases observed with larger, more difficult and unstable antibodies, such as scFV.

To address the reviewer's concerns, we have changed the text in the discussion to clarify:

“Using this scaffold in an *in vitro* evolution screen, we were able to generate novel quenchbodies for the detection of IL-6 with 1.5–2.4 fold-sense. This is a marked improvement over the fold-sense of 1.3–1.5 obtained for the quenchbodies we generated as models against MBP and lysozyme in the present study. While higher fluorescence fold-increases have been reported for larger antibody- or ScFV-based quenchbodies [11, 27-29, 38], or for nanobody-based quenchbodies that detect small molecules such as methotrexate [18, 19], other nanobody-based quenchbodies that detect purified proteins only report fluorescence fold-increases of 1.2–1.4 [19, 39].”

I confirmed the correction in the explanation of Fig. 3E. However, I do not agree with the conclusion that all three Trp residues are involved in quenching based on the result that the W-to-A triple mutant of anti-MBP showed no response, without examining the binding activity by pull-down. This result cannot exclude the possibility that the W-to-A triple mutant has lost binding activity and therefore the Trp residues involved in quenching cannot be identified, and the possibility that only one or two of the three Trp residues are involved in quenching.

We agree with the reviewer that some tryptophans may be more or less important than others, but since the tryptophans are involved in both binding and quenching, there is no way to test this experimentally and arrive at a definitive conclusion. This is why our biochemical experiments are complemented by *in silico* predictions, which further support the importance of all three tryptophans in the quenching mechanism (Fig. S1A). To further clarify these points, we have added the following text to the end of the results section titled “CDR-tryptophans underpin the quenchbody sensing mechanism”:

“Formally, our experimental results do not disentangle loss of antigen binding and loss of quenching. Instead, our results report on the overall quenchbody performance, which depends on both antigen binding and quenching by the tryptophans, with the relative importance of each tryptophan further considered using complementary *in silico* modelling. Based on this information, tryptophans that contributed favourably to sensing performance across the lysozyme and MBP quenchbodies were considered to be W59, W101, W103, W110, and W115.”

I appreciate that the authors conducted additional experiments with W-to-Y mutants of Variants #1, #15, and #36 in accordance with my comments. However, the authors carried out the similar inappropriate experimental plan. The result that the W-to-Y quintuple mutant showed no response does not exclude the possibility that the binding activity was lost and therefore the quenching effect of Trp residues cannot be examined, and the possibility that only some Trp residues are involved in quenching and other Trp residues are worthless.

Similar to our response to our previous comment, we have now included text in at the end of the results section titled “CDR-tryptophans underpin the quenchbody sensing mechanism” (as above), which addresses potential limitations of our experimental plan.

We reiterate that our experimental results do not distinguish between antigen binding and quenching, but instead focus on quenchbody performance, which is a product of both antigen binding and quenching. We have also added the following text to the discussion to acknowledge the relative usefulness of different tryptophans, to address the concerns of the reviewer:

“We speculate that some IL-6 quenchbodies failed to respond or had lower responses, because IL-6 is a relatively small antigen, and depending on the mode of binding, it is possible that tryptophans are accessible to the fluorophore at the antigen–quenchbody interface if IL-6 does not completely envelope the convex-binding surface of the quenchbodies. Therefore, some tryptophans may be redundant for successful performance, and higher quenchbody performance may be possible through further engineering of library-derived quenchbodies to remove redundant tryptophans. Alternatively, evolution of convex-binding nanobodies containing conserved CDR-tryptophans to bind large protein targets which completely occupy the convex-binding surface might lead to higher fluorescence performance for future quenchbody development.”

I agree the usefulness of Quenchbody libraries containing Trp residues that may be involved in quenching, and I commend the authors for investigating the usefulness. However, I consider this manuscript insufficient to be accepted for three reasons: the result in Fig. 3E does not provide evidence that W101, W110, and W115 should be included in the library; the effects of individual Trp in the anti-IL6 variants have not been examined; and the improvement in response is not significant.

We have added text to address all the points highlighted by the reviewer. See our responses to the comments above.